# Recovery of Hydrochloric Acid from Industrial Wastewater by Diffusion Dialysis Using a Spiral-Wound Module

**DOI:** 10.3390/ijms23116212

**Published:** 2022-06-01

**Authors:** Arthur Merkel, Ladislav Čopák, Daniil Golubenko, Lukáš Dvořák, Matej Vavro, Andrey Yaroslavtsev, Libor Šeda

**Affiliations:** 1MemBrain s. r. o. (Membrane Innovation Centre), Pod Vinicí 87, 471 27 Stráž pod Ralskem, Czech Republic; matej.vavro@membrain.cz (M.V.); libor.seda@membrain.cz (L.Š.); 2Institute for Nanomaterials, Advanced Technologies and Innovation, Technical University of Liberec, Studentská 2, 461 17 Liberec, Czech Republic; lukas.dvorak@tul.cz; 3Kurnakov Institute of General and Inorganic Chemistry of the Russian Academy of Sciences, 31 Leninsky Avenue, 119991 Moscow, Russia; golubenkodaniel@yandex.ru (D.G.); yaroslav@igic.ras.ru (A.Y.)

**Keywords:** acid recovery, anion-exchange homogeneous membrane, soluble salts, metallic chlorides, membrane stability, techno-economic study, wastewater

## Abstract

In the present study, the possibility of using a spiral-wound diffusion dialysis module was studied for the separation of hydrochloric acid and Zn^2+^, Ni^2+^, Cr^3+^, and Fe^2+^ salts. Diffusion dialysis recovered 68% of free HCl from the spent pickling solution contaminated with heavy-metal-ion salts. A higher volumetric flowrate of the stripping medium recovered a more significant portion of free acid, namely, 77%. Transition metals (Fe, Ni, Cr) apart from Zn were rejected by >85%. Low retention of Zn (35%) relates to the diffusion of negatively charged chloro complexes through the anion-exchange membrane. The mechanical and transport properties of dialysis FAD-PET membrane under accelerated degradation conditions was investigated. Long-term tests coupled with the economic study have verified that diffusion dialysis is a suitable method for the treatment of spent acids, the salts of which are well soluble in water. Calculations predict significant annual OPEX savings, approximately up to 58%, favouring diffusion dialysis for implementation into wastewater management.

## 1. Introduction

Acid regeneration from industrial wastewater containing salts of Zn, Cu, Fe, Ni, Co, and other elements using diffusion dialysis is an efficient, affordable, and environmentally friendly process [1,2,3,4]. The key element of a diffusion-dialysis based acid-recovery system is an anion-exchange membrane, which has a high acid permeability and can retain metal cations [5]. In recent years, the synthesis of new membranes and the modification of known materials are the main directions in the development of DD [6,7,8,9], together with module structure optimization [10,11] and researching new applications [12,13,14,15]. Currently, “plate-and-frame” is the most commonly used membrane module geometry; however, the space-saving characteristics and modular nature of spiral-wound DD membrane modules have attracted much attention [4,11,16]. Previously, we conducted a study on the dependence of volumetric flow and feed temperature on the performance of the same spiral-wound module for the recovery of free sulfuric acid from 5.4% wt. sulfuric acid solution containing 284 ppm of Fe^2+^ and 50 ppm of Cu^2+^ [17]. The results indicated that an increase in the volumetric flow rate of water increased free acid yield from 88% to 93%, but decreased Cu^2+^ and Fe^2+^ ion rejection from 95% to 90% and from 91% to 86%, respectively. It was also found that the membranes changed their properties during operation within a month: a slight drop in permselectivity and an increase in acid and salt permeability. Obtained results showed that the payback time of the membrane-based regeneration unit is approximately within one year.

Membrane stability is a critical factor for their use in DD, which characterizes the steadiness of the membrane permeability and selectivity under operation conditions [4]. Increasing attention is paid to the stability of anion exchange membranes in alkaline media [18,19,20] due to their promising use in alkaline fuel cells and electrolysis modules. In contrast, their stability in acids is understudied.

In the present study, we assess the performance and economic efficiency of the DD spiral-wound module to separate hydrochloric acid and Zn^2+^, Ni^2+^, Cr^3+^, and Fe^2+^ salts. In addition, membrane degradation through the characterization of ion-exchange capacity and mechanical and transport properties before and after stress tests was explored. Particularly, the impact of volumetric flow was investigated.

## 2. Results

### 2.1. Effect of Volumetric Flow Ratio

Based on the conductivity and flowrate record, the steady state of Test 1 (flow rate of both streams regulated by 9 L∙h^−1^ flow restrictors) of the module performance was reached 2 h after turning on the dialysis unit (Figure 1). During the run-up, no pressure fluctuations resulting in issues with maintaining the demanded flowrate were observed. Conductivities increasing over time (Figure 1a) indicate the saturation of the diffusate stream, mainly with HCl. In the case of the dialysate stream, high conductivity values result from the presence of the acid and salt.

The values of the last three points of mass flowrate are accurate enough to calculate the performance parameters of Test 1. The inlet flowrate of the filtered feed was 155.1 g∙min^−1^. The output flowrates of 114.4 g∙min^−1^ for the diffusate and 190.7 g∙min^−1^ for the dialysate were obtained. Taking the flowrate values and sample analyses into evaluation, the process characteristics of Test 1 were estimated and are summarized in Table 1. Despite achieving a steady performance, the HCl yield ratio is slightly above 68%. A low rejection factor of Zn^2+^ was also observed. Furthermore, Ni^2+^ and Fe^2+^ are retained by an AEM equally by 85%.

A small portion of Cr^3+^ (ca. 5%) passed through the membrane into a diffusate chamber. Due to the significant diffusion of Zn^2+^ into the diffusate, the purity of the regenerated acid stream is 13.1%. Analytical measurements also revealed that the waste stream contained only 2.6% wt.; the HCl is twice as much less pure as the obtained product stream. The concentration of the HCl (9.5% wt.) in the product stream is very close to the initial value in the feed stream (10.3% wt.).

The increase of the volumetric flow rate of the stripping medium (Test 2; volumetric flow rate of the tap water was regulated by the 12 L∙h^−1^ flow restrictor) led to the creation of a new concentration profile of the output streams (Figure 2). The lower electrical conductivity of the diffusate stream (582 mS·cm^−1^) indicates a lower acid concentration.

At the inlet flowrate of the tap water, 200 g∙min^−^^1^ and 143.4 g∙min^−1^ of the filtered feed, 150.4 g∙min^−1^ of the diffusate, and 193.0 g∙min^−1^ of the dialysate were obtained. Performance parameters calculated from the flowrates and the samples’ composition are in Table 2.

Most of the fed Zn^2+^ (65%) cations diffused through the membrane into the diffusate compartment. For Test 2, free acid is saved by 77%. The remaining 23% of HCl leaves the membrane module as a waste stream altogether with non-complexing metals, which are rejected by more than 90%. In terms of stream composition (Table 3), the waste stream is approximately 2.1 times less pure than the product stream (the degree of purity is 13%).

The concentration of the free HCl in the recovered stream (7.6% wt.) is quite far from the concentration of HCl in the inlet filtered feed. However, acid concentration in the outgoing stream is almost five times less than in the incoming stream.

### 2.2. Membrane Properties

The manufacturer’s module includes a commercial anion exchange membrane Fumasep^®^ FAD-PET-75 (Fumatech BWT GmbH, Bietigheim-Bissingen, Germany). According to technical specifications, Fumasep^®^ FAD-PET-75 is a PET-reinforced anion exchange membrane with low resistance, high acid transfer rate, high mechanical stability, and high stability in an acidic environment. FAD-PET-75 membrane ageing was carried out by its treatment during three weeks in the feed solution of 25 °C and 60 °C. Membrane ageing in 25 °C solution leads to a 6% decrease in tensile strength and an elongation at break increase of 42%. After ageing in the 60 °C solution, a threefold drop in tensile strength and elongation at break was observed (Figure 3b, Table 4). At the same time, there are no significant differences in the IR spectra (Figure 3a).

Visually, there are no cracks or traces of degradation of the polyethylene terephthalate (PET) mesh on both SEM images (Figure 4). Stability tests almost did not change the ion exchange capacity of the membranes but significantly affected their water uptake, thickness, ionic conductivity, and permselectivity (Table 4).

The effect of stability tests on membrane acid permeability and acid/metal separation selectivity was studied in batch dialysis experiments (Figure 5 and Appendix A). The average acid flux through the initial membrane was 1.34 mmol·h^−1^·cm^−2^ and increased for membranes subjected to ageing up to 2.60 in 25 °C solution and up to 3.51 in 60 °C solutions. The dependence of the acids’ and metals’ concentration in the dialysate chamber versus time during the batch dialysis of the feed solution is close to linear. The acid dialysis coefficient was 4.41·10^−3^ mol·m^−2^·h^−1^ for membrane before use and increased up to 8.33·10^−3^ mol·m^−2^·h^−1^ and 1.13·10^−2^ mol·m^−2^·h^−1^ after a stability test at 25 °C and 60 °C, respectively (Table 5). The obtained values of the acid dialysis coefficient are close to the previously published values for Fumatech membranes [21]. The metal dialysis coefficients for the membrane after a 60 °C stability test are much higher than after a 25 °C stability test and before use. The selectivity coefficient for Ni, Cr, and Fe separation varies within 30–150, which is close to the best dialysis with commercial and tailor-made membranes [8,9,22]; however, the zinc separation selectivity is low—the *S* factor is close to 1. The Ni, Cr, and Fe selectivity factors after the 60 °C stability test are lower than after the 25 °C stability test.

### 2.3. Economic Analysis of Acid Recovery System

A fictional plant consuming 15% wt. HCl solution for production purposes and generating 10% wt. HCl wastewater containing transition metals has been used to evaluate the economic study. An economic analysis of the implementation of the DD recovery system uses the results obtained from Test 1 (volumetric flowrates of the inlet streams are regulated by the 9 L·h^−1^ flow restrictors). A HCl-resistant unit consisting of 10 spiral-wound WD-AR10-2001 modules was used for evaluation. Annual expenses and savings are related to 330 working days of the 24/7 regime. Two cases of wastewater management were considered (Figure 6).

Case (A) considers the neutralization of the spent acid, separating the precipitated hydroxides after adding the technical-grade 90% Ca(OH)_2_. The moisture of the precipitated hydroxides was 60%. The obtained filtrate was diluted with the tap water at the concentration of the dissolved inorganic solids to 3000 ppm. Case (B) considers implementing the DD system after the production process and the treatment of the dialysate waste stream in the same way as in Case A. Pre-treatment of the spent acid prior to the DD in the form of filtration through a 10 μm cartridge was considered. Pre-treatment expenses (cartridge replacement frequency) were estimated from the testing experience (1 cartridge per 2 m^3^ of the processed raw liquid). Both cases consider the preparation of the fresh 15% wt. HCl stream consumed in the production process.

The unit costs of the required media (electrical energy, tap water, technical-grade slaked lime, solid waste disposal expenses, concentrated 31% wt. HCl) are summarized in Table 6.

Solving the mass balance of the proposed technological scheme coupled with data obtained from Test 1 (recovery ratio of HCl 68%; rejection of Fe 84%, Ni 85%, Zn 39%) resulted in an annual cost demand summarized in Table 7. 

A comparison of the annual expenses has shown significant improvements in every situation, regardless of the minor demand in water consumption (as a stripping medium), the electricity demand (for powering the pumps of the recovery unit), and the pre-treatment of the spent acid. Annual savings reach 58% of OPEX for the case without implementing the membrane-based recovery system.

## 3. Discussion

Compared to the result of Test 1, the conductivity of the diffusate stream at the moment of sampling (582 mS·cm^−1^) is lower than the conductivity of the diffusate stream of Test 1 (662 mS·cm^−1^). This comparison leads to a statement that a higher volumetric flowrate of tap water impacts the system performance. The higher flowrate of water than that of the spent acid into the module yields more acid. However, the ratio of free acid is not as high as expected (ca. 90%). The reason for this might be the significant passage of chlorides in the form of Zn chloro complexes and the osmosis of water. Due to the high osmotic pressure of the dialysate, the osmosis of water takes place from the diffusate stream to the dialysate stream, which is the opposite direction to the acid diffusion. The opposite flow of water decreases the acid diffusion rate through the AEM. The acid concentration may change due to an interaction with bicarbonate ions present in tap water. However, their concentration in tap water is three orders of magnitude lower than the acid concentration, and cannot significantly change its value.

The retention of Zn^2+^ in continuous-regime experiments and the separation factor in batch experiments is low due to the formation of chloro complexes. The fraction of each complex ([ZnCl]^+^; ZnCl_2_; [ZnCl_3_]^−^; [ZnCl_4_]^2−^) depends on a concentration of HCl (Figure 7a). At a high concentration of HCl in the solution, dissolved Zn is mainly in the form of negatively charged chloro complexes (Figure 7b), which is a case of the processed spent HCl solution. In the solution containing 115 g·L^−1^ HCl, approximately 84% of the dissolved Zn is negatively charged. The uncharged Zn complex can also diffuse through a thin membrane due to a non-zero concentration gradient. However, at the end of the module-active area, the local rejection factor of Zn will not be as low as at the beginning of the module-active area due to a low concentration of free HCl (28 g·L^−1^). The dominant Zn chloro complex in the solution containing 28 g·L^−1^ of HCl is ZnCl_2_ with [ZnCl]^+^. In general, according to the graph of the total charge of Zn species (Figure 7b), the solution containing more than 33 g·L^−1^ of HCl is prone to a low rejection rate of the dissolved Zn. The negative effect of zinc chloride complexes on the efficiency of the diffusion dialysis and the strong effect dependence on the HCl concentration were also noted in [12].

The doubly charged ions of the iron triad elements, in contrast to zinc, are not prone to the formation of chloride complexes. The most stable [FeCl]^+^ complex forms iron. However, its stability constant is very low and amounts to only 2.3 [23]. 

Changes in the properties of membranes after their stability tests should also be discussed (Table 4). Keeping them in the used feed solution leads to tensile strength and permselectivity decreases, water uptake, and conductivity increases. This is especially evident when tested at elevated temperatures (60 °C). It should be noted that these changes correlate with those occurring during membrane treatment at an elevated temperature and water activity. As described in [24], this is determined by an increase in the pore volume due to an increase in the water uptake of the membranes. In this case, an increase in the size of pores and channels takes place, which limits the conductivity of membranes [25]. At the same time, the volume of the electrically neutral solution in the center of the pores also increases, which leads to a decrease in permselectivity [26,27,28]. In addition, water entering the pores of the membranes plays the role of a plasticizer, and therefore, an increase in water uptake leads to a decrease in the strength of the membranes [29].

The observed increase in acid permeability correlates with ionic conductivity change and is related to increased water uptake (Table 5). However, separation factors for the membrane after a 25 °C stability test are more than for the initial membrane, although membrane permselectivity decreases. According to the dialysis-coefficient change, the observed improvement in selectivity-coefficients is mainly due to a sharp increase in acid permeability. It is known that protons are transported by a unique mechanism (Grotthuss mechanism) which consists in the diffusion of “excess” protons through the hydrogen bond network of water molecules [25,29]. Apparently, an increase in the water uptake of membranes after the stability tests leads to a more mobile network of hydrogen bonds, ensuring efficient proton transfer. The authors of [9] also observed the passage of the selectivity of dialysis membranes through a maximum with a change in the degree of hydration.

However, not only the proton permeability increased after a 25 °C stability test; the permeability of zinc and iron ions also increases, by 74% and 45%, respectively, while the permeability of nickel and chromium ions decreases by 33% and 23%, respectively (Table 5). This permeability change led to the increase in the separation selectivity for nickel and chromium more than for iron, while for zinc, the separation selectivity remains almost unchanged. In general, the increase in the ions’ permeability can be explained by membrane water uptake growth, which leads to an increase in the mobility of all particles. However, the drop in permeability for nickel and chromium is not clear. These data correlate with the stability of chloride complexes. For the zinc, all complexes up to ZnCl_4_^2−^ are known; for iron FeCl^+^ and FeCl_2_, and for nickel and chromium, only MeCl^+^ with small stability constants is known [23,30]. Furthermore, in the case of iron, the possibility of oxidation with the formation of Fe^3+^, which also forms relatively stable chloride complexes, cannot be ruled out [31]. This behaviour of the ions’ permeability is probably the result of a complex relationship between the form of the ions in the solution, the interaction of these forms with the functional groups of the membrane, and diffusion in the membrane matrix. After the 60 °C stability test, the permeability of all elements increased, but for metal ions, it was higher than for protons; therefore, the separation selectivity for all elements decreased.

From the material structure point of view, an increase in membrane swelling is probably associated with the conformational mobility of polymer chain fragments, which increases with increasing temperature. This leads to the pore and channel systems’ restructuring.

The temperatures for stability tests were chosen assuming that keeping the membrane in the 25 °C feed solution will give us an understanding of what properties the membrane will have shortly after the start of operation of the module. Exposure at elevated temperature (60 °C) was carried out for accelerated degradation to evaluate material properties by the end of its operational life. Thus, the acid permeability through the FAD-PET-75 membrane will increase during operation, while the selectivity will pass through a maximum. Moreover, the deterioration of the module’s efficiency with operation can occur due to cracks in the membranes, due to changes in their volume and to membrane degradation. This effect is noticeable in electron micrographs of membranes subjected to prolonged testing at elevated temperatures.

The economic analysis of a generic plant was based on the results obtained from the test, where the volumetric flows of the inlet streams were regulated by the 9 L·h^−1^ flow restrictors. The higher flowrate of the stripping medium improves the yield of the HCl. However, this configuration leads to the production of larger volumes of recovered acid. Moreover, the HCl solution used to prepare the fresh 15% wt. stream already bears much water (compared to 96% wt. H_2_SO_4_, which is more concentrated). To avoid bringing too much water into the system, the results of Test 2 were excluded from the evaluation. Nevertheless, a low rejection of Zn can increase its steady concentration in the production-process bath over a safe concentration level. A high Zn concentration in the bath solution can be decreased by implementing the second stage of dialysis, adding an inlet stream of fresh water, or considering the higher flowrate of the stripping medium.

After the neutralization and filter press, dilution with tap water was considered for the filtrate’s post-treatment. As an alternative for dilution, thermal regeneration is known. However, thermal methods are energy demanding. In cases where the amount of produced metal-chloride-rich solutions is enormous, thermal regeneration is not an ideal option. A factor not considered was the presence of side streams (their pH, salinity) typical for any production process. For this reason, any economic viability of the proposed technology should be assessed individually, based on the data obtained from the measurements of the representative sample and information from the qualified personnel.

## 4. Material and Methods

### 4.1. Feed Solution

Wastewater used for testing purposes was provided by the company CVP Galvanika s.r.o. (Ždánice, Czech Republic). The spent acid solution sample came from the ZnFe and ZnNi electroplating process. The solution had no visible signs of oil and was filtered through a 10 μm cartridge prior to acid recovery to remove solid particles from the liquid. The composition of the processed feed is summarized in Table 8.

The stripping medium used for HCl recovery was tap water. The selected quality parameters are in Table 9. 

### 4.2. Analytical Methods

The determination of the composition was undertaken via potentiometric alkalimetry for measuring the free acid content, and inductively coupled plasma (ICP-OES), coupled with optical emission spectroscopy (both Thermo Fischer Scientific GmbH, Bremen, Germany), was used for measuring the metallic ion concentration by the method of a calibration curve. Standard samples had concentrations of 0.1, 0.5, 1; 5, 10, and 100 ppm. Chloride anions were measured using the isotachophoresis method on an Agrofor device (JZD Odra, Krmelín, Czech Republic).

The main parameters measured during the acid-recovery experiments were electrical conductivity, density, volumetric flow, and mass flow. Electrical conductivity was determined using a TetraCon 925/LV-P probe (Xylem Analytics WTW, Weilheim, Germany) connected to a WTW 3430 Multimeter (Xylem Analytics WTW, Germany). Density was measured with a portable hand-held Densito 30PX density meter (Mettler Toledo, Chiyoda, Japan). A KERN 572 balance (Kern & Sohn GmbH, Balingen, Germany), cylinder, and a timer were used to determine the mass flow.

### 4.3. Equipment

A Heidolph PD 5001 peristaltic pump, (Schwabach, Germany) and a 10 μm cartridge (České filtry s.r.o., Ústí nad Labem, Czech Republic) were used to pre-treat the spent HCl solution.

A WD-AR10-2001 spiral-wound module (Spiraltec GmbH, Sachsenheim, Germany) with an effective anion exchange area of ca. 5 m^2^ was used to test the effect of the flow rate ratio on the recovery of spent HCl by DD. 

### 4.4. Membrane Characterization

AC-membrane ionic conductivity was determined via the four-electrode method, using a thermostatic 25 °C cell filled with a 0.5 M NaCl solution and a P-40X potentiostat/galvanostat with an FRA-24M impedance measurement module (Elins, Chernogolovka, Russia). A detailed description of the experiment can be found elsewhere [32].

Permselectivity was characterized by the potentiostat technique, the membrane being placed between 0.1 M and 0.5 M NaCl solutions in a two-compartment cell. A detailed description of the experiment and calculations can be found elsewhere [32]. Stress–strain experiments were performed using a Tinius Olsen H5KT universal testing machine and a force sensor set at 100 N under ambient conditions (24 °C/25% relative humidity). The gauge length of the samples was adjusted to 50 mm for the machine and the strain rate was set at 5 mm/min for all tests. A more detailed description of the experiment can be found elsewhere [33]. Water uptake was determined from weight loss after drying the film at 80 °C for several hours. Dimensional swelling was determined by the change in the geometric dimensions of the membrane sample before and after dehydration. Scanning electron microscope (SEM) images were obtained using a Quanta FEG 450 SEM (FEI, Hillsboro, OR, USA) at 10 kV accelerating voltage and 80 Pa residual pressure. A detailed description of the ion exchange capacity measurement and calculation can be found elsewhere [34]. The thickness of each sample was taken as the average value of five points measured before the experiment using a Mitutoyo 293–805 micrometer (Mitutoyo, Sakado, Japan). FTIR spectra of the samples were measured using a Nicolet iS5 spectrometer (Thermo Fisher Scientific, Waltham, MA, USA) in attenuated total reflection mode using a Quest Specac accessory (500–4000 cm^−1^ spectral range, 32 scans, 2 cm^−1^ resolution). The OMNIC^®^ IR software Advanced ATR Correction was used to correct the resulting spectra (bounces number—1, angle of incidence—45°, refractive index—1.5).

Batch-dialysis experiments were conducted at room temperature (24 °C) in a two-section PTFE cell with a 5 cm^2^ active membrane area. The feed solution volume was 25 mL; the dialysate solution volume was 30 mL. A tailor-made two-position magnetic stirrer constantly stirred the solutions at 400 pm.

### 4.5. Diffusion Dialysis Tests

The industrial sample was pre-treated with a filtration process before entering the recovery system. When a sufficient amount of filtered feed was collected in a tank, the recovery unit was turned on. When the cartridge inserted into a translucent housing became visibly dirty, it was replaced with a new cartridge. The process is visualized in Figure 8.

Two different volumetric flowrates (9 L·h^−1^ and 12 L·h^−1^) of the stripping medium were chosen to study the system performance at fixed volumetric flowrate (8 L·h^−1^ ± 1 L·h^−1^) of the spent acid. Tests were performed at the ambient temperature of 23 °C–27 °C. Different volumetric flowrates of the tap water were set by 9 L·h^−1^ and 12 L·h^−1^ flow restrictors. Both tests were performed in a one-pass regime until a steady performance was reached. After de-airing the module, samples of the diffusate and the dialysate of each test used to evaluate the system performance were taken under steady conditions. The module was flushed with tap water before the first test (Test 1). Upon taking the samples of Test 1, the flow restrictor for the water circuit was changed to 12 L·h^−1^ (Test 2). The recovery unit was in a working regime during the night. Samples of the output streams of Test 2 were also taken after de-airing the module. 

### 4.6. Calculations

The DD process in a continuous regime was characterized by equations of acid recovery ratio, metal ion rejection, and purities of the obtained streams, which were proposed in a previous work [17].

A distribution diagram of Zn species was obtained by solving a system of non-linear equations describing the Zn–ligand equilibria in solution and the mass balance of Zn and Cl:(1)$${\mathrm{Zn}}^{2+}+{\mathrm{Cl}}^{-}⇄{\left[\mathrm{ZnCl}\right]}^{+};\;\beta_{1}=\frac{\left[{\left[\mathrm{ZnCl}\right]}^{+}\right]}{\left[{\mathrm{Zn}}^{2+}\right]\cdot \left[{\mathrm{Cl}}^{-}\right]}=2.7$$
(2)$${\mathrm{Zn}}^{2+}+2{\mathrm{Cl}}^{-}⇄{\mathrm{ZnCl}}_{2}\;;\;\beta_{2}=\frac{\left[{\mathrm{ZnCl}}_{2}\right]}{\left[{\mathrm{Zn}}^{2+}\right]\cdot {\left[{\mathrm{Cl}}^{-}\right]}^{2}}=4.1$$
(3)$${\mathrm{Zn}}^{2+}+3{\mathrm{Cl}}^{-}⇄{\left[{\mathrm{ZnCl}}_{3}\right]}^{-};\;\beta_{3}=\frac{\left[{\left[{\mathrm{ZnCl}}_{3}\right]}^{-}\right]}{\left[{\mathrm{Zn}}^{2+}\right]\cdot {\left[{\mathrm{Cl}}^{-}\right]}^{3}}=3.4$$
(4)$${\mathrm{Zn}}^{2+}+4{\mathrm{Cl}}^{-}⇄{\left[{\mathrm{ZnCl}}_{4}\right]}^{2-};\;\beta_{4}=\frac{\left[{\left[{\mathrm{ZnCl}}_{4}\right]}^{2-}\right]}{\left[{\mathrm{Zn}}^{2+}\right]\cdot {\left[{\mathrm{Cl}}^{-}\right]}^{4}}=1.6$$
(5)$$c_{0,\mathrm{Zn}}=\left[{\mathrm{Zn}}^{2+}\right]+\left[{\left[\mathrm{ZnCl}\right]}^{+}\right]+\left[{\mathrm{ZnCl}}_{2}\right]+\left[{\left[{\mathrm{ZnCl}}_{3}\right]}^{-}\right]+\left[{\left[{\mathrm{ZnCl}}_{4}\right]}^{2-}\right]$$
(6)$$c_{0,Cl}=\left[{\mathrm{Cl}}^{-}\right]+\left[{\left[\mathrm{ZnCl}\right]}^{+}\right]+2\cdot \left[{\mathrm{ZnCl}}_{2}\right]+3\cdot \left[{\left[{\mathrm{ZnCl}}_{3}\right]}^{-}\right]+4\cdot \left[{\left[{\mathrm{ZnCl}}_{4}\right]}^{2-}\right]$$
where $\left[{\left[{\mathrm{ZnCl}}_{x}\right]}^{2-x}\right]$ is an equilibrium molar concentration of Zn species (*x*
∈
$\left\{0;\;1;\;2;\;3;\;4\right\}$), $\left[{\mathrm{Cl}}^{-}\right]$ is an equilibrium molar concentration of chlorides, and *c*_0_ is an analytical molar concentration of Zn (Cl). Values of the overall formation constants ***β*** of Zn^2+^ complexes were adopted from [23].

Water uptake (*W*) was calculated by the following equation (Equation (7)):(7)$$W=\frac{m_{\mathrm{wet}}-m_{\mathrm{dry}}}{m_{\mathrm{dry}}}\cdot 100\%$$
where *m*_wet_ and *m*_dry_ represent the membrane weight before and after dehydration, respectively.

The dialysis coefficients (*U*) of each component were calculated as follows:(8)$$U=\frac{M}{S\cdot t\cdot \Delta C}$$
where *M* represents the number of moles of transported components determined from the concentration dependence of components in the dialysate chamber on time, *S* is the effective membrane area in m^2^, and t denotes the test time in hours. Δ*C* is the logarithmic average of the concentration difference between two chambers, defined as (9):(9)$$\Delta C=\frac{C_{f}^{0}-C_{d}^{t}-C_{f}^{t}}{\ln\left[\left(C_{f}^{0}-C_{d}^{t}\right)/C_{f}^{t}\right]}$$
where $C_{f}^{0}$ and $C_{f}^{t}$ are the concentration of the individual component in the feed at times 0 and *t*, respectively, and $C_{d}^{t}$ is the concentration of the component in the dialysate at time t. As the dialysis degree was low (<10%), the ΔC was assumed to be close to $C_{f}^{0}$.

The acid/metal *X* separation factor (*S*_H/X_) was determined based on the dialysis coefficient *U*_H_/*U*_X_ ratio.

### 4.7. Statistical Analysis

Measurement uncertainties for electrical conductivity and temperature were obtained from the manufacturer’s documentation. In contrast, measurement uncertainties for heavy metal ion concentration, chlorides, and acidity were obtained by applying the Student t-distribution with a significance level of *p* < 0.05. The mass flow was characterized by a measurement uncertainty of 2%, obtained from the propagation-of-error formula.

## 5. Conclusions

The treatment of the industrial sample by diffusion dialysis with a spiral-wound membrane module has shown that HCl acid can be recovered by 68% (77%) depending on the experimental configuration. The higher the stripping-medium flowrate, the higher the recovery ratio and the more diluted streams are obtained. Due to the complexing nature of Zn with chloride anions, the retention of Zn is low, approximately 35%. The rejection factor of Fe, Ni, and Cr is approximately more than 85%, depending on the cation charge. FAD-PET membrane after accelerated degradation at 60 °C has lower strength and separation selectivity and greater acid permeability. 

A techno-economic analysis has shown that diffusion dialysis decreases annual OPEX costs by 58%. The post-treatment of the proposed technology scheme consists of dialysate neutralization, the separation of the transition-metal hydroxides, and the dilution of the chloride-rich filtrate to 3 g·L^−1^ of dissolved solids. The last, conventional step might be a bottleneck of the scheme since water becomes increasingly scarce. Therefore, attention should be paid to developing more elaborate treatment methods of chloride-salt-rich solutions and the extraction of other input materials (Zn, Ni).

## Figures and Tables

**Figure 1 ijms-23-06212-f001:**
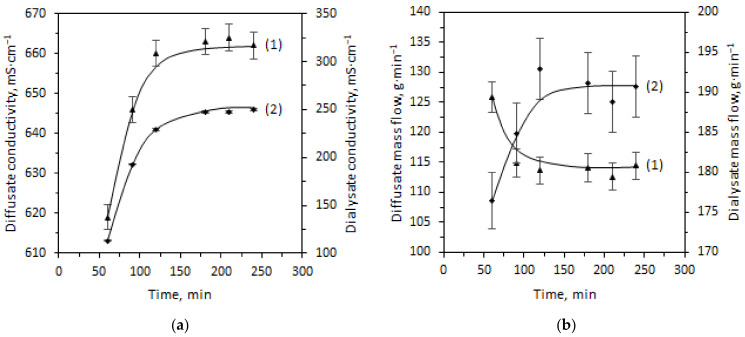
Electrical conductivity (**a**) and mass flowrate (**b**) of the output streams for Test 1 over time; (1) (▲) diffusate, (2) (◆) dialysate.

**Figure 2 ijms-23-06212-f002:**
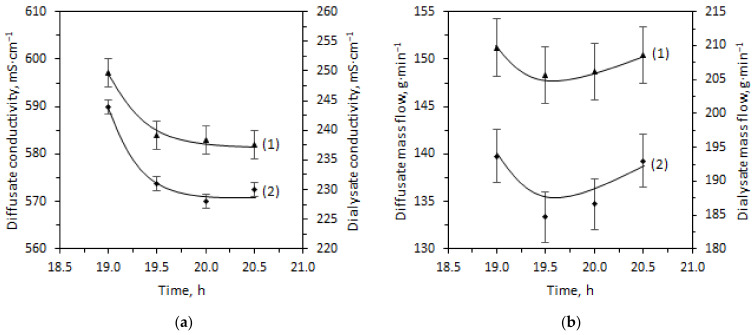
Electrical conductivity (**a**) and mass flowrate (**b**) of the output streams for Test 2; (1) (▲) diffusate, (2) (◆) dialysate.

**Figure 3 ijms-23-06212-f003:**
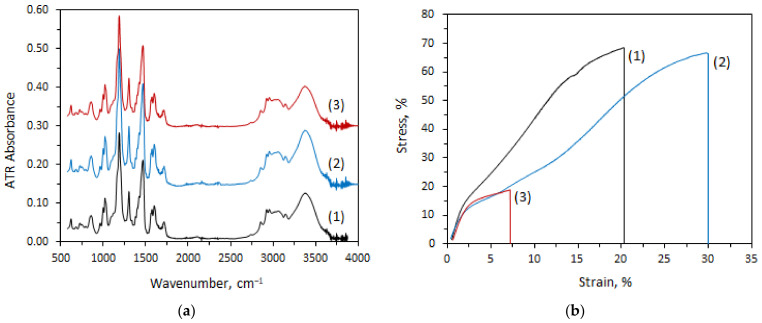
Fourier transform infrared spectroscopy (FT-IR) spectra (**a**) and (**b**) stress-strain curves for membranes (1) before use (initial state) and after stress-tests at 25 °C (2) and 60 °C (3) (in the air-dry state, Cl^−^ form).

**Figure 4 ijms-23-06212-f004:**
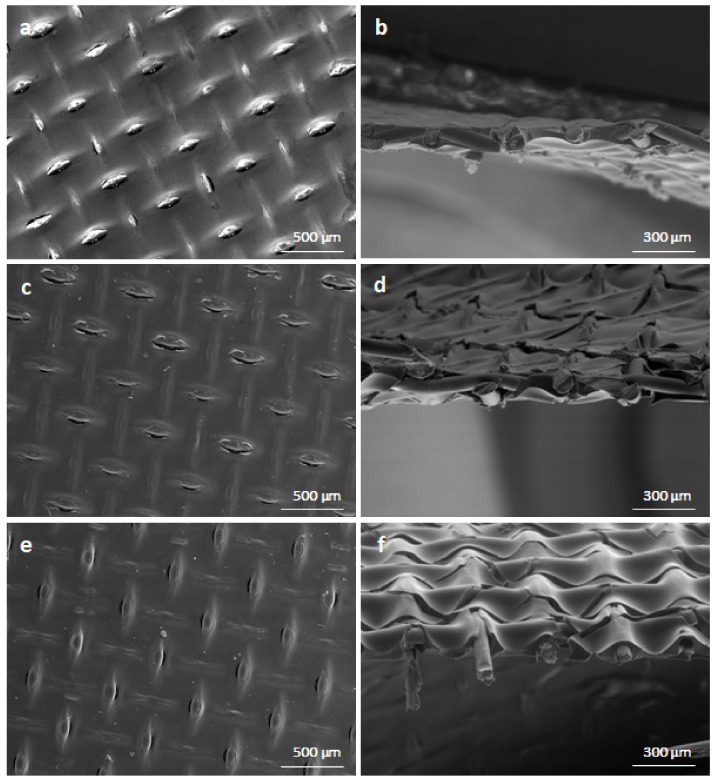
Scanning electron microscopy images detailing membrane state before use ((**a**) surface, (**b**) cross-section)) and after use (ageing at 25 °C: (**c**) surface, (**d**) cross-section; ageing at 60 °C: (**e**) surface, (**f**) cross-section).

**Figure 5 ijms-23-06212-f005:**
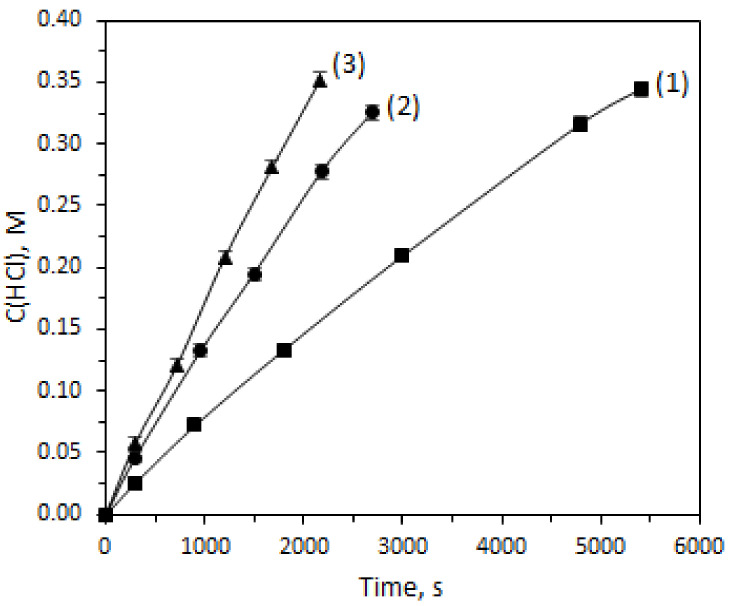
The HCl concentration in the dialysate chamber versus time during batch dialysis of feed solution for membranes (1) (∎) before use (initial state) and after stability tests in 25 °C (2) (•) and 60 °C (3) (▲) solutions. The concentration of metals in the dialysate chamber versus time during batch dialysis of the feed solution is given in the Appendix A.

**Figure 6 ijms-23-06212-f006:**
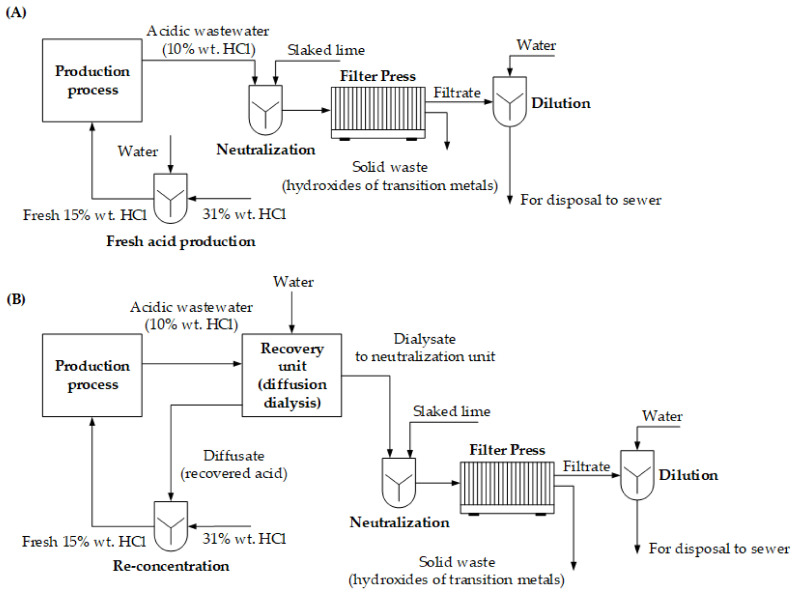
Proposed technology for HCl-rich wastewater treatment. (**A**) Treatment of acidic wastewater without diffusion dialysis; (**B**) implementation of diffusion dialysis into production-level wastewater management.

**Figure 7 ijms-23-06212-f007:**
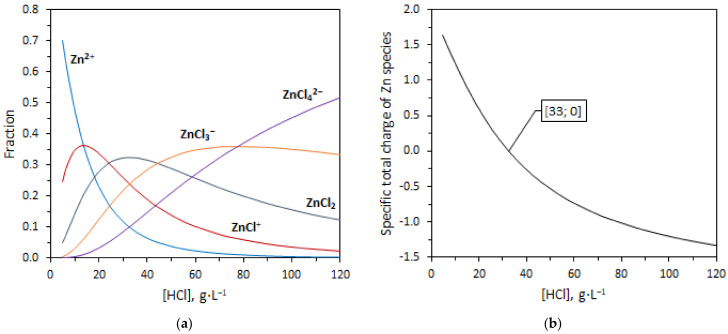
Distribution of Zn species in HCl solution (**a**) and (**b**) specific charge of Zn species as a function of the concentration of free HCl. Fraction values were obtained by numerically solving Equations (1)–(6).

**Figure 8 ijms-23-06212-f008:**
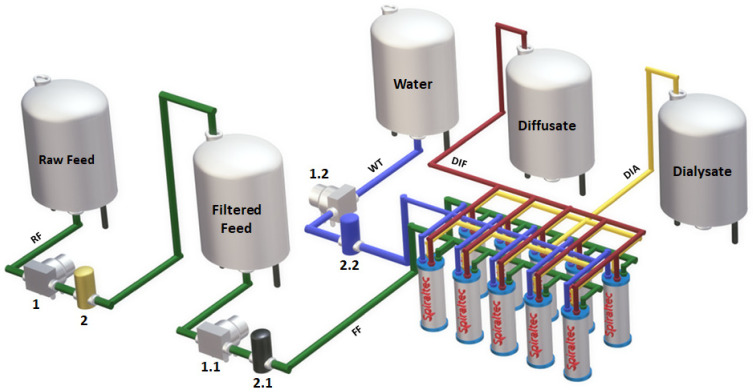
Visualization of the acid-recovery system. Instrumentation key: (1; 1.1; 1.2) pumps; (2; 2.1; 2.2) filters.

**Table 1 ijms-23-06212-t001:** Performance parameters of Test 1.

Parameter	Unit	Value
HCl yield	%	68 ± 5
Cr^3+^ rejection	%	95 ± 2
Fe^2+^ rejection	%	84 ± 3
Ni^2+^ rejection	%	85 ± 3
Zn^2+^ rejection	%	39 ± 15

**Table 2 ijms-23-06212-t002:** Performance parameters of Test 2.

Parameter	Unit	Value
HCl yield	%	77 ± 6
Cr^3+^ rejection	%	95 ± 1
Fe^2+^ rejection	%	92 ± 2
Ni^2+^ rejection	%	94 ± 2
Zn^2+^ rejection	%	35 ± 17

**Table 3 ijms-23-06212-t003:** Analytical results of the output streams for Test 1 and Test 2.

Composition	Unit	Feed	Diffusate	Dialysate	Diffusate	Dialysate
			Test 1		Test 2	
HCl	g∙L^−^^1^	114 ± 2	103 ± 2	27.4 ± 0.5	81 ± 1	25.3 ± 0.5
	% wt.	10.3 ± 0.2	9.5 ± 0.2	2.6 ± 0.1	7.6 ± 0.2	2.4 ± 0.1
Fe^2+^	ppm	5500 ± 500	810 ± 80	216 ± 20	350 ± 30	3500 ± 300
Zn^2+^	ppm	22,600 ± 2300	17,900 ± 1800	6500 ± 600	14,500 ± 1500	6200 ± 600
Ni^2+^	ppm	1440 ± 100	216 ± 20	900 ± 90	94 ± 4	950 ± 90
Cr^3+^	ppm	37 ± 4	3.6 ± 0.1	28 ± 3	1.6 ± 0.2	27 ± 3
Density	kg∙L^−1^	1.108 ± 0.001	1.083 ± 0.001	1.039 ± 0.001	1.064 ± 0.001	1.037 ± 0.001

**Table 4 ijms-23-06212-t004:** Physical and chemical parameters of Fumasep^®^ FAD-75 in Cl^−^ form.

Parameter	Unit	Value (Initial)	Value (25 °C)	Value (60 °C)
Thickness (wet)	μm	78 ± 1	88 ± 1	93 ± 1
Ion exchange capacity	meq·g^−1^	1.55 ± 0.02	1.50 ± 0.02	1.51 ± 0.02
Specific conductivity in 0.5 M NaCl	mS·cm^−1^	11.9 ± 0.6	18 ± 1	24.5 ± 2
Permselectivity (at 0.1/0.5 mol·kg^−1^ NaCl)	%	95.2 ± 0.4	93.0 ± 0.4	91.3 ± 0.4
Water uptake	% wt.	23.5 ± 0.5	36.2 ± 1.0	37.0 ± 1.0
Young’s modulus	MPa	1160 ± 70	780 ± 34	890 ± 27
Yield strength	MPa	13.1 ± 1.0	11.4 ± 0.4	13.7 ± 1.5
Tensile strength	MPa	68.5 ± 0.7	64.6 ± 1.4	21.4 ± 2
Elongation at break	%	20.0 ± 1.0	28.5 ± 1.6	8.5 ± 1.2

**Table 5 ijms-23-06212-t005:** Batch tests’ dialysis coefficients (*U*) and separation factors (*S*).

Membrane	*U_H_*	*U_Fe_*	*U_Zn_*	*U_Ni_*	*U_Cr_*	*S_H_* _/_ * _Fe_ *	*S_H_* _/_ * _Zn_ *	*S_H_* _/_ * _Ni_ *	*S_H_* _/_ * _Cr_ *
Original	4.4 ± 0.1·10^−3^	1.1 ± 0.1·10^−4^	4.7 ± 0.2·10^−3^	1.2 ± 0.1·10^−4^	6.0 ± 0.8·10^−5^	39 ± 3	0.93 ± 0.04	36 ± 3	74 ± 9
Stability test at 25 °C	8.3 ± 0.2 ·10^−3^	1.6 ± 0.1·10^−4^	8.2 ± 0.3·10^−3^	0.8 ± 1·10^−4^	4.6 ± 0.1·10^−5^	52 ± 3	1.02 ± 0.03	101 ± 13	181 ± 6
Stability test at 60 °C	1.13 ± 0.02·10^−2^	3.4 ± 0.4·10^−4^	10.0 ± 0.1·10^−3^	2.7 ± 0.4·10^−4^	7.9 ± 0.4·10^−5^	34 ± 4	1.13 ± 0.07	42 ± 1	144 ± 7

**Table 6 ijms-23-06212-t006:** Unit costs of the used media, including waste disposal. The prices of the listed items reflect the situation in the Czech Republic.

Item	Unit	Unit Cost
Electricity	EUR/kWh	0.1
Tap water	EUR/m^3^	2.8
31% wt. HCl	EUR/metric ton	140.0
90% Ca(OH)_2_	EUR/metric ton	138.8
Solid waste disposal Cartridge 10 μm	EUR/metric ton EUR/pc	83.2 6.4

**Table 7 ijms-23-06212-t007:** Annual expenses for each case of the economic study. (**A**) Processing spent 10% wt. HCl without DD; (**B**) processing spent 10% wt. HCl with DD.

Case	Unit	A	B	Savings
Water for DD	EUR	0	1996	−1996
Water for TDS dilution	EUR	141,176	51,085	+90,091
Water for bath preparation	EUR	1033	0	+1033
Solid waste disposal	EUR	6303	3155	+3148
Electricity for DD unit (including pumps for filtration)	EUR	0	1584	−1584
Pre-treatment (cartridges)	EUR	0	2128	−2128
31% wt. HCl	EUR	48,438	23,997	+24,441
Technical Ca(OH)_2_	EUR	15,593	5663	+9930
Total	EUR	212,543	89,608	+122,935

**Table 8 ijms-23-06212-t008:** Composition of the feed sample.

Composition	Unit	Feed Stream
Electrical conductivity	mS·cm^−1^	658 ± 3
Density	kg·L^−1^	1.108 ± 0.001
HCl	g·L^−1^	114 ± 2
Cr^3+^	ppm	37 ± 4
Fe^2+^	ppm	5470 ± 500
Ni^2+^	ppm	1440 ± 100
Zn^2+^	ppm	22,600 ± 2300
Cl^−^	ppm	128,200 ± 6400

**Table 9 ijms-23-06212-t009:** The average composition of the feed solution.

Parameter	Unit	Value
Electrical conductivity	mS·cm^−1^	0.632 ± 0.003
pH	-	7.58 ± 0.01
Hardness (Ca^2+^ + Mg^2+^)	mmol·L^−1^	2.6 ± 0.1
Alkalinity (pH 4.5)	mmol·L^−1^	3.75 ± 0.08
SO_4_^2−^	ppm	111 ± 5
Cl^−^	ppm	25 ± 1

## Data Availability

The data presented in this study are available on request from the corresponding author.

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
