# Peer review of "Recovery of Hydrochloric Acid from Industrial Wastewater by Diffusion Dialysis Using a Spiral-Wound Module"

_ijms, 2022, doi:10.3390/ijms23116212_

Round 1

Reviewer 1 Report

The authors studied  the possibility of using a spiral-wound diffusion dialysis module  for the separation of hydrochloric acid and Zn2+, Ni2+, Cr3+ and Fe2+ salts. Diffusion dialysis recovered 68% of free HCl.  When a higher volumetric flowrate was used  recovered a 77%  free acid was recovered.  Transition metals (Fe, Ni, Cr) apart from Zn were rejected by >85%. Low retention of Zn (35%) relates to the diffusion of negatively charged chloro complexes through the anion-exchange membrane. Techno-economic analysis has shown that diffusion dialysis decreases annual OPEX costs by 58%. This paper is a continuation of a previously work done by authors in which they studied the dependence of volumetric flow and feed temperature on the performance of the same spiral wound module for the recovery of free sulfuric acid containing  Fe2+ and  Cu2+ . The paper is well structured, the results are clear and discussions are consistent. Schemes are intuitive.

Author Response

Response to Reviewer 1

Point: The authors studied the possibility of using a spiral-wound diffusion dialysis module for the separation of hydrochloric acid and Zn2+, Ni2+, Cr3+ and Fe2+ salts. Diffusion dialysis recovered 68% of free HCl.  When a higher volumetric flowrate was used  recovered a 77%  free acid was recovered.  Transition metals (Fe, Ni, Cr) apart from Zn were rejected by >85%. Low retention of Zn (35%) relates to the diffusion of negatively charged chloro complexes through the anion-exchange membrane. Techno-economic analysis has shown that diffusion dialysis decreases annual OPEX costs by 58%. This paper is a continuation of a previously work done by authors in which they studied the dependence of volumetric flow and feed temperature on the performance of the same spiral wound module for the recovery of free sulfuric acid containing Fe2+ and  Cu2+. The paper is well structured, the results are clear and discussions are consistent. Schemes are intuitive.

Response: Dear reviewer, thank you very much. We appreciate this positive comment.

Reviewer 2 Report

The study is well designed and nicely presented. However, I think it will benefit rom some revisions listed below.

Figure 1, could you please explain such high uncertainties for the mass flowrate?

Figure 1, caption, the meaning of each symbol (star, triangle) must be described.

Figure 5 from SI should be named Figure S1 instead.

Lines 204-218, Wouldn’t it be possible to add some ions that form very stable complexes with Zn(II) to achieve better results?

Table 3, why the values for Feed are provided twice since they are the same? This doesn’t increase the clarity and is just duplication of the data.

Figure 3, FT-IR, why the Authors have not done the baseline correction?

Lines 366-370, exactly how this system has been solved and using what kind of software?

Author Response

Response to Reviewer 2

The study is well designed and nicely presented. However, I think it will benefit from some revisions listed below.

The authors thank the reviewer for the comments that help us to clarify some questions and improve the manuscript. Please find our response to the comments of the reviewer marked in red and changes in the manuscript marked in blue.

Point 1: Figure 1, could you please explain such high uncertainties for the mass flowrate?

Response 1: High uncertainties for the mass flowrate are caused by air which got into the module. During an operation of the unit, we observed flow of air bubbles from the module in the tubes of the outlet streams. We had to de-air the system several times before we took the samples. This is visible even more in Figure 2b when even after 19 hours of operation, a significant drop in flowrates occurred.  

Point 2: Figure 1, caption, the meaning of each symbol (star, triangle) must be described.

Response 2: Changes in Figure 1a,b; Figure 2a,b and Figure 5 has been implemented. Moreover, indication of diffusate is (1) and dialysate is (2).

Point 3: Figure 5 from SI should be named Figure S1 instead.

Response 3: Change has been implemented.

Point 4: Table 3, why the values for Feed are provided twice since they are the same? This doesn’t increase the clarity and is just duplication of the data.

Response 4: Change has been implemented.

Point 5: Lines 204-218, Wouldn’t it be possible to add some ions that form very stable complexes with Zn(II) to achieve better results?

Response 5:  Thank you very much for this point. This is a really good point. Probably, adding EDTA (H4Y) into a raw feed (which pH is very close to 0) will result into formation of ZnH2Y which has zero charge. Nevertheless, some diffusion will be observed anyway. This idea can be used in future cases as HNO3/HF contaminated with Fe.

Point 6: Figure 3, FT-IR, why the Authors have not done the baseline correction?

Response 6: The spectra were modified using the advanced ATR correction whose description was added to the experimental part.

Point 7: Lines 366-370, exactly how this system has been solved and using what kind of software?

Response 7:  The system of algebraic equations 1 – 6 was solved numerically in Excel by using Solver. An objective function F (set to a minimum) was defined as follows: 

(Calculation is attached in the Word file)

β values are constants. Equilibrium concentrations of Zn2+, ZnCl+, ZnCl2, ZnCl3-, ZnCl42-, Cl- are changing variables. Analytical concentrations of Zn and Cl had to be chosen to satisfy the condition of a zero degree of freedom: we have 6 independent equations, thus very 6 variables can be computed from the equation system. The constraint was a sum of the fraction of each Zn species equal to 1. Method used for solving the system of 6 equations (consisting of both linear and strongly non-linear algebraic equations) was a gradient method.

For chosen values of analytical concentrations of HCl and Zn (in form of ZnCl2), Solver was run. Obtained curves of Zn species distribution are smooth because computational procedure was performed for a lot of points, especially at extremes (maximums, inflex points) of the curves.

Reviewer 3 Report

This work used a spiral-wound diffusion dialysis module to recover acid using the diffusion dialysis method. The mechanical and transport properties of commercial FAD-PET membranes under accelerated degradation were investigated in detail. The topic is important and the work is well-organized. The research is well-designed and the results are well-presented. I think this work can be published in IJMS.

The FAD-PET-75 is a PET (Polyester) reinforced AEMs. It is interesting that the PET mesh could be seen clearly at the surface of the membrane (Figure 4a). After use, there is some wrinkle on the ion-exchanged phases, and the surface smoothness of FAD-PET-75 becomes worse. Maybe the author can give some discussion.

In Table. 5, the SH/Ni and SH/Fe of the original membrane are similar while the SH/Ni is twice as much as SH/Fe when the membrane was tested at 60 . Please give some discussion.

Some novel AEMs used for acid recovery have been developed and the author may read and discuss them in the paper, such as Chem Eng J 2022, 445, 136752.

Author Response

Response to Reviewer 3

This work used a spiral-wound diffusion dialysis module to recover acid using the diffusion dialysis method. The mechanical and transport properties of commercial FAD-PET membranes under accelerated degradation were investigated in detail. The topic is important and the work is well-organized. The research is well-designed and the results are well-presented. I think this work can be published in IJMS.

The authors thank the reviewer for the comments that help us to clarify some questions and improve the manuscript. Please find our response to the comments of the reviewer marked in red and changes in the manuscript marked in green.

Point 1: The FAD-PET-75 is a PET (Polyester) reinforced AEMs. It is interesting that the PET mesh could be seen clearly at the surface of the membrane (Figure 4a). After use, there is some wrinkle on the ion-exchanged phases, and the surface smoothness of FAD-PET-75 becomes worse. Maybe the author can give some discussion.

Response 1: As it turned out, different sides of the FAD-PET membranes have different morphology (Fig 1). On one side, the reinforcing mesh comes through the polymer, and it is smoother, and on the other, it does not show through, but it is wavier. Therefore, we obtained image of the other side of the original membrane and replaced the Figure. This illustrates that there is no significant effect of degradation on membrane morphology.

(SEM images attached in the Word file)

a

b

Fig 1. SEM images of different sides of the original FAD-PET membrane

In the case of cross-section photographs, for different membranes, a different viewing angle, and as it turned out, different sides have different morphology; in addition, sample preparation in liquid nitrogen leads to cracking of the material. Unfortunately, fine effects such as wrinkle and smoothness cannot be discussed in connection with the above.

Point 2: In Table. 5, the SH/Ni and SH/Fe of the original membrane are similar while the SH/Ni is twice as much as SH/Fe when the membrane was tested at 60 ℃. Please give some discussion. 

Response 2: After 60 °C stability test, the selectivity coefficients of Ni and Fe are close. Apparently the reviewer has in mind a significant difference in the selectivity of these metals after 25 °C stability test. This is because the permeability for Fe ions increases, and for Ni ions decreases. The permeability for zinc also increases but decreases for chromium. In general, the increase of ions permeability can be explained by membrane water uptake growth, which leads to an increase in the mobility of all particles. However, the drop in permeability for nickel and chromium is not clear. These data correlate with the stability of chloride complexes. So for zinc, all complexes up to ZnCl42− are known, for iron FeCl+ and FeCl2, and for nickel and chromium only MeCl+ with small stability constants [23, D.F.C. Morris, G.L. Reed, E.L. Short, D.N. Slater, D.N. Waters, Nickel (II) chloride complexes in aqueous solution, J. Inorg. Nucl. Chem. 27 (1965) 377–382. doi:10.1016/0022-1902(65)80355-4. D.F.C. Morris, G.L. Reed, E.L. Short, D.N. Slater, D.N. Waters, Nickel (II) chloride complexes in aqueous solution, J. Inorg. Nucl. Chem. 27 (1965) 377–382. doi:10.1016/0022-1902(65)80355-4]. Also, in the case of iron, the possibility of oxidation with the formation of Fe3+, which also forms relatively stable chloride complexes, cannot be ruled out. [I. Persson, Ferric Chloride Complexes in Aqueous Solution: An EXAFS Study, J. Solution Chem. 47 (2018) 797–805. doi:10.1007/s10953-018-0756-6.]. This behaviour of ions’ permeability is probably the result of a complex relationship between the form of ions in the solution, the interaction of these forms with the functional groups of the membrane, and diffusion in the membrane matrix.

We gave more discussion on it in the Discussion part.

Point 3: Some novel AEMs used for acid recovery have been developed and the author may read and discuss them in the paper, such as Chem Eng J 2022, 445, 136752.

Response 3: Thank you for the relevant citation; it has been added to the manuscript.

Round 2

Reviewer 2 Report

The Authors have revised and improved their manuscript. Current version can surely be accepted for publication.